# Modern Surgery-First Approach Concept in Cleft-Orthognathic Surgery: A Comparative Cohort Study with 3D Quantitative Analysis of Surgical-Occlusion Setup

**DOI:** 10.3390/jcm8122116

**Published:** 2019-12-02

**Authors:** Hyung Joon Seo, Rafael Denadai, Betty Chien-Jung Pai, Lun-Jou Lo

**Affiliations:** 1Department of Plastic and Reconstructive Surgery, Craniofacial Research Center, Chang Gung Memorial Hospital, Chang Gung University, Taoyuan City 33302, Taiwan; payton100@hanmail.net (H.J.S.); denadai.rafael@hotmail.com (R.D.); 2Department of Plastic and Reconstructive Surgery and Biomedical Research Institute, Pusan National University Hospital, Busan City 49241, Korea; 3Division of Craniofacial Orthodontics, Department of Dentistry, Chang Gung Memorial Hospital, Taoyuan City 33302, Taiwan

**Keywords:** occlusion setup, orthognathic surgery, surgery first, modified surgery first, cleft lip/palate

## Abstract

Despite the evident benefits of the modern surgery-first orthognathic surgery approach (reduced treatment time, efficient tooth decompensation, and early improvement in facial esthetics), the challenge of the surgical-occlusion setup acts as a hindering factor for the widespread and global adoption of this therapeutic modality, especially for the management of cleft-skeletofacial deformity. This is the first study to assess three-dimensional (3D) quantitative data of the surgical-occlusion setup in surgery-first cleft-orthognathic surgery. This comparative retrospective study was performed on 3D image datasets from consecutive patients with skeletal Class III deformity who had a unilateral cleft lip/palate (cleft cohort, *n* = 44) or a noncleft dentofacial deformity (noncleft cohort, *n* = 22) and underwent 3D computer-assisted single-splint two-jaw surgery by a single multidisciplinary team between 2014 and 2018. They received conventional orthodontics-first or surgery-first approaches. 3D quantitative characterization (linear, angular, and positional measurements) of the final surgical-occlusion setup was performed and adopted for comparative analyses. In the cleft cohort, the occlusion setup in the surgery-first approach had a significantly (all *p* < 0.05) smaller number of anterior teeth contacts and larger incisor overjet compared to the conventional approach. Considering the surgery-first approach, the cleft cohort presented significantly (all *p* < 0.05) larger (canine lateral overjet parameter) and smaller (incisor overjet, maxillary intercanine distance, maxillary intermolar distance, ratio of intercanine distance, and ratio of intermolar distance parameters) values than the noncleft cohort. This study contributes to the literature by providing 3D quantitative data of the surgical-occlusion setup in surgery-first cleft-orthognathic surgery, and delivers information that may assist multidisciplinary teams to adopt the surgery-first concept to optimize cleft care.

## 1. Introduction

Cleft lip, with or without a cleft palate, is the second most common global birth defect, affecting 1.7 in every 1000 births [1]. A substantial percentage of skeletally mature patients (20–75%) [2,3,4] present visible skeletofacial deformities and cleft-related psychosocial problems that contribute to social discrimination and stigmatization [5,6], requiring appropriate treatment [2,3,4]. Orthognathic surgery (OGS) is considered the mainstay of treatment due to the positive impact on oral function, facial appearance, and psychosocial health [7,8,9]. Classically, the dental arches of patients with clefts have been orthodontically prepared before OGS [10,11]. This conventional orthognathic pathway (or orthodontics-first approach) involves complete dental management, including 7–47 months of presurgical therapy for correction of dental compensation, arch alignment, maxillary-and mandibular-arch coordination, and the leveling of accentuated occlusal-plane discrepancies [11,12,13]. However, this therapeutic method has been associated with prolonged overall treatment time [12,13,14].

A modern orthognathic pathway (or surgery-first approach) has recently been introduced into the armamentarium of multidisciplinary teams treating dentoskeletofacial deformities [15,16,17]. Surgical intervention preceding orthodontic treatment has evident advantages over the conventional orthodontics-first approach, including immediate postoperative improvement of facial appearance and a substantial reduction of total treatment time [18,19,20]. However, the rate of surgery-first approach-related complications may be slightly higher than those associated with orthodontics-first approach [21], which should be considered by multidisciplinary teams delivering orthognathic care.

In this setting, published guidelines have directed multidisciplinary teams during the decision-making process of selecting patients for the surgery-first approach [22,23]. Most surgery-first-related studies have reported patients with Class III malocclusion with no associated congenital anomaly [18,19,20], with cleft-related deformity being considered as an exclusion criterion for the surgery-first approach [24]. Patients with repaired cleft lip and palate have a significant aesthetic impairment [25] and the main benefit of surgery-first approach is an immediate increase of the quality of life after surgery, due to the improvement of facial appearance [26].

Recommendations have also been published to direct the surgical-occlusion setup (i.e., the setup of the transitional occlusion at the time of surgery), the most difficult step for the surgery-first approach [22,23,27,28,29,30,31,32,33,34,35,36,37,38]. However, most of these previous recommendations had no quantitative data or occlusal characteristics [27,28,29,30,31]. Existing occlusal-related quantitative data are restricted to the surgery-first approach in noncleft skeletal Class III deformity [38]. We are not aware of any study focused on quantitative analysis of the surgical-occlusion setup in surgery-first cleft-orthognathic surgery. The setting of surgical occlusion is paramount to orthognathic therapeutic flow as it serves not only to anticipate the dental movements necessary in postsurgical orthodontic treatment, but also to fabricate the occlusal splint that guides surgeons during operations. Therefore, definition of quantitative data for occlusal-related parameters from a three-dimensional (3D) perspective can support understanding the modern surgery-first pathway concept in the context of cleft-skeletofacial deformity and also inspire the expansion of its use among multidisciplinary centers delivering cleft care.

The purposes of this study were: (1) to assess the 3D quantitative characteristics of the surgical-occlusion setup in surgery-first cleft-orthognathic surgery; and (2) to compare these data with those obtained from conventional cleft-orthognathic-surgery and surgery-first noncleft orthognathic-surgery approaches. 3D quantitative-based recommendations for the surgical-occlusion setup in surgery-first cleft-orthognathic surgery were also addressed.

A primary outcome and a secondary outcome are expected in this study:

The primary endpoint of the study is the comparison of surgical-occlusion setup between surgery-first (experimental group) and conventional orthognathic surgery (control group) in cleft patients/cohort.

The secondary endpoint of the study is the comparison of surgical-occlusion setup between cleft and noncleft patients/cohorts.

The null hypotheses were:

No difference of surgical-occlusion setup exists between surgery-first (experimental group) and conventional orthognathic surgery (control group) in cleft patients/cohort.

No difference of surgical-occlusion setup exists between cleft and noncleft patients/cohorts.

## 2. Material and Methods

The Strengthening the Reporting of Observational Studies in Epidemiology (STROBE) guidelines were used for reporting the results of this institutional review board–approved (Chang Gung Medical Foundation, protocol 201900008B0) comparative retrospective study. Consecutive skeletally mature patients (finished their growth spurt with no more increase in body height) with Class III occlusion were recruited who had unilateral complete cleft lip/palate (cleft cohort) or noncleft deformity (noncleft cohort) undergoing orthodontic-surgical treatment by the Chang Gung multidisciplinary team between 2014 and 2018. All included patients were treated by the same orthodontist (B.C.-J.P.) and surgeon (L.-J.L.). Demographics (age and gender), orthodontic-surgical (dental characteristics, type of orthognathic approach, time of presurgical therapy, and need for revision bone and/or soft tissue surgery to improve occlusal, maxillary, mandibular, and/or chin morphology within the follow-up), and cone beam computed tomography (CBCT) scan data were retrieved from the Craniofacial Research Center database.

Exclusion criteria were patients with Class I or II skeletal patterns; bilateral cleft deformity; any syndromic diagnosis; previous orthognathic surgery; and/or an incomplete medical/3D image record. Patients who underwent segmental osteotomies (surgically assisted maxillary expansion, anterior subapical osteotomy, and/or segmented maxillary osteotomies) were also excluded.

Sixty-six patients with cleft (*n* = 44; 50% female; 19 ± 4 (15–23) years at surgery; 66% with left-sided clefts) and noncleft (*n* = 22; 50% female; 22 ± 5 (15–27) years at surgery) deformities met the described criteria.

### 2.1. Orthognathic-Surgery Treatment

Full descriptions of the pre- and postorthognathic-treatment principles used in this center were previously published [22,23,28,32,33,34,35,36,37,38,39,40,41,42,43]. After a short period of orthodontic preparation for secondary alveolar bone grafting (9-year-old patients), no further surgical intervention or orthodontic treatment have been performed in patients with unilateral cleft lip/palate who present signs of skeletofacial deformity such as Class III skeletal pattern. The orthognathic surgery treatment process has been started when the patients reach the skeletal maturity (>15 and 18 years for female and male patients, respectively).

All included patients were surgically treated by using 3D computer-assisted single-splint two-jaw orthognathic surgery [38,39,40,41]. Selected patients were orthodontically managed by using conventional orthodontics-first or surgery-first approaches [22,23,28,32,33,34,35,36,37] based on dentition status at presentation:
(a)In the conventional approach, surgery was performed after a period of at least seven months of complete orthodontic therapy, including the leveling and alignment of dental arches to eliminate any occlusal interference at surgery, and the removal of all dental compensations to maximize optimal surgical repositioning of the jaws.(b)Patients with different compositions of the dental conditions (Table 1) received the surgery-first orthognathic treatment based on a patient-specific therapeutic planning (Figure 1). Patients with more and less favorable dental conditions received the surgery-first treatment as it was based on the orthodontist’s judgment of achievement of a practicable surgical-occlusion setup as well as anticipation of a feasible postoperative orthodontic treatment. Our team stratified the surgery-first approach into two models. In the standard surgery-first model, surgery was performed with no need for presurgical orthodontic therapy. In the modified surgery-first model, a short period (≤6 months) of orthodontic therapy was performed preoperatively. This presurgical dental adjustment was exclusively implemented for the reduction of potential dental collisions and the minimal decompression of mandibular teeth.

In this study, the 3D quantitative data of standard and modified surgery-first models were initially compared. Both models were then compiled as a unique dataset (surgery-first approach) and adopted for comparative analysis between surgery-first and conventional approaches.

### 2.2. 3D-Image Acquisition

One month before surgery, a standard craniofacial CBCT scan was performed for each patient using an i-CAT 3D Dental Imaging System (Imaging Sciences International, Hatfield, PA, USA) with the following parameters: 120 kVp, 0.4 mm × 0.4 mm × 0.4 mm voxel size, 40 s scan time, and a 22 cm × 16 cm field of view. The patients’ head was positioned with the Frankfort horizontal plane parallel to the ground. Throughout the scan, patients were instructed not to swallow, to keep their mouth closed, and to maintain a centric occlusion bite [32,33,38,40]. Images were stored in Digital Imaging and Communications in the Medicine format and rendered into 3D volumetric images using the Dolphin 3D software package (Dolphin Imaging and Management Solutions, Chatsworth, CA, USA).

### 2.3. Final Surgical-Occlusion Setup

Two weeks before surgery, the final surgical-occlusion setup was manually performed using the dental-cast model method, considering dental midline coincidence, canting, and the relative position of dentitions between maxillary and mandibular arches. The target was to avoid severe postoperative occlusal instability, incomplete or excessive skeletal correction, or skeletofacial deformities such as asymmetry while defining the surgical-occlusion setup.

To transfer the dental-cast method into a digital image, the maxillary and mandibular dental casts and the defined surgical occlusion were scanned (Figure 2) by using a surface scanner (3-Shape, Copenhagen, Denmark). Using Dolphin software, the dentition in CBCT was superimposed and replaced by a digitalized dental image [42,43]. The 3D skull models were oriented according to the Frankfort horizontal and midsagittal planes. Osteotomy lines were created by segmenting the maxilla (Le Fort I) and mandible (bilateral sagittal split osteotomy). The digitalized dental image was then manipulated by moving the osteotomized distal mandible segment to the fixed maxilla for achievement of the desired final surgical occlusion (Figure 3, Figure 4 and Figure 5).

### 2.4. Virtual Planning

One week before surgery, 3D computer-assisted surgical planning was performed by the surgeon and orthodontist. To achieve a normal jaw relationship with skeletofacial harmony and symmetry, the maxillomandibular complex was mobilized while preserving the surgical-occlusion setup (Figure 6) [38,40]. If any modification in the surgical-occlusion setup was judged as necessary (e.g., surgical unfeasibility due to inappropriate initial occlusion setup), the overall process was redesigned (including a new final surgical-occlusion setup) until consensus was achieved between surgeon and orthodontist. The fabrication of computer-generated 3D surgical-occlusion splints (Figure 7) was accomplished by only adopting the final surgical-occlusion setup as template for adjustment of thickness (OrthoAnalyzer software package; 3Shape, Copenhagen, Denmark) and printing (Objet30 OrthoDesk 3D Printer, Stratasys Ltd., Rehovot, Israel). These 3D-printed final surgical-occlusion splints were adopted in the surgical procedures. No intermediate surgical-occlusion setup or intermediate occlusal splint was adopted in the authors’ approach.

### 2.5. Type of Orthognathic Approach

The cleft cohort had 23 (52%) and 21 (48%) patients who underwent conventional orthodontics-first and surgery-first approaches, respectively. Eight (38%) and 13 (62%) patients received the standard and modified surgery-first models, respectively.

The noncleft cohort had 4 (18%) and 18 (82%) patients who underwent the conventional orthodontics-first and surgery-first approaches, respectively. Eleven (61%) and seven (39%) patients received the standard and modified surgery-first models, respectively.

### 2.6. Surgical Approach

All the included patients received 3D computer-assisted single-splint two-jaw orthognathic surgery (final occlusal splint, 1-piece Le Fort 1 maxillary osteotomy, and bilateral sagittal split osteotomy) according to the previously described by our team [38,39,40]. Using the 3D simulated image as a guiding template (Figure 8), the maxillomandibular complex with 3D-printed final surgical-occlusion splint was moved to the desired position (Figure 9). The Le Fort I was initially fixed by using 2-mm titanium miniplates placed on the nasomaxillary and zygomatico-maxillary pillars bilaterally, with no rigid fixation in the anterior maxillary walls. Three-hole miniplates and 6 mm screws were routinely bent to match the maxillary contour at the Le Fort I osteotomy line, ensuring the desired position of the maxillomandibular complex. Longer miniplates, i.e., four or five holes, were alternatively employed to overcome potential drawbacks related to the presence of weak maxillary bone or osteotomy-induced fracture in the medial or lateral maxillary pillar region [39]. After Le Fort I fixation, the proximal ramus segment was placed in a relaxed position and gently pushed up to ensure the position of the condylar head in the glenoid fossa. Percutaneous insertion of three bicortical screws 14–16 mm long was performed in the ramus. No interpositional bone graft was used. Intermaxillary fixation was released and the occlusion was evaluated. Genioplasty was finally executed as planned, along with intraoperative judgement. The patients with no intermaxillary fixation were admitted in regular ward for two days following the surgery and then clinically examined based on regular surgical and orthodontic appointments. A liquid diet was recommended in the first week, followed by a soft diet in the second week.

### 2.7. 3D Quantitative Analysis

The 3D image datasets displaying the final surgical-occlusion setup adopted for surgery were included for analysis as they represented the occlusion setup in the context of surgical feasibility. 3D quantitative analyses of occlusion characteristics were performed based on dentoskeletofacial parameters defined in a previous investigation [32]: dental-occlusion contacts (number and location), overjet/overbite, angle molar relation (Class I, II, or III), posterior open bite, transverse arch coordination, dental inclination, midline and transverse discrepancies, and jaw relationship (ANB angle and A-point–nasion–B-point angle). All 3D image datasets were analyzed by an investigator with no information about the type of orthodontic approach by using Dolphin software tools (line, angle, and occlusogram with a color map). Twenty randomly selected patients’ CBCT scans were measured in duplicate, with one-month interval between each measurement.

Accuracy of surgical occlusion was determined by assessing the number of occlusions requiring two setups [32]. The 3D CBCT-derived cephalometric normative data for the Taiwanese Chinese population were adopted as the reference value of the jaw relationship (ANB angle = 3.3 ± 1.6 (0.5–6.2) degrees [44]).

### 2.8. Statistical Analysis

In descriptive analysis, data were presented as means ± standard deviations. It was verified that the data were normally distributed by using the Kolmogorov–Smirnov test. The Student t-test and chi-square test were used for the comparative analyses. Two-sided values of *p* < 0.05 were considered statistically significant. All analyses were performed using SPSS Version 17.0 (IBM Corp., Armonk, NY, USA).

## 3. Results

No new surgical-occlusion setup was required during virtual planning. All patients were treated by two-jaw orthognathic surgery, with no intraoperative problems with the 3D-printed final-occlusion splints. On average, a normal jaw relationship was noticed after virtual planning in the cleft and noncleft cohorts (ANB angle = 3.4 and 3.2 degrees, respectively; *p* > 0.05). Three patients with cleft and one patient with noncleft presented with lip or chin numbness at 1–6 months postoperatively, with full recovery at long-term evaluations. No wound infection, postoperative hemorrhage/hematoma, or requirement or request for revisionary surgery during follow-up was observed in the cleft and noncleft cohorts.

### 3.1. Time of Presurgical Orthodontic Therapy

The time for presurgical orthodontic therapy was similar between cohorts, with 11.7 ± 3.8 and 10.2 ± 7.4 (*p* = 0.447) months in the conventional orthodontics-first approach, and 4.9 ± 1.6 and 4.0 ± 1.0 (*p* = 0.171) months in the modified surgery-first model for the cleft and noncleft cohorts, respectively. Patients who underwent the standard surgery-first model had no presurgical orthodontic therapy.

### 3.2. Primary Endpoint

No significant difference was observed in comparison between standard and modified surgery-first models for all parameters (Table 2).

There were significant (all *p* < 0.05) differences in the comparison between surgery-first and conventional orthodontics-first approaches in the number of anterior-tooth contacts and incisor overjet parameters, with no significant difference for the remainder of the tested parameters (Table 3).

Comparative analyses considering the cleft side revealed significantly (all *p* < 0.05) larger (interincisal angle, U1 overjet, and presence of posterior open-bite parameters) and smaller (number of anterior-tooth contacts and U1 inclination parameters) values for the surgery-first and conventional approaches, respectively. Considering the noncleft side, the surgery-first approach had significantly (all *p* < 0.05) larger (interincisal angle, U1 overjet, and presence of posterior open-bite parameters) values than conventional approach (Table 4).

### 3.3. Secondary Endpoint

Considering the surgery-first approach, the cleft cohort presented significantly (all *p* < 0.05) larger (canine lateral overjet parameter) and smaller (incisor overjet, maxillary intercanine distance, maxillary intermolar distance, ratio of intercanine distance, and ratio of intermolar distance parameters) values than the noncleft cohort (Table 5).

### 3.4. Reliability

Intra-investigator reliability was considered excellent (intraclass correlation coefficients = 0.898 to 0.975) for all quantitative parameters.

## 4. Discussion

In this comparative study of occlusion setup, the primary endpoint-related data releveled a smaller number of anterior teeth contacts and larger incisor overjet in the surgery-first cleft-orthognathic surgery approach than the conventional cleft-orthognathic surgery approach, which clinically represents an incisor decompensation postponed after surgery. Patients with cleft also had smaller overjet and higher anterior contacts in cleft side with conventional orthognathic surgery than surgery first approach, which characterizes the typical status of dentition in surgery first-treated patients who had the upper incisors positioned in a more upright position due to surgical procedure-derived scar contraction. Moreover, the secondary endpoint-related data demonstrated a larger canine lateral overjet and smaller incisor overjet and maxillary transversal-related distances in the surgery-first cleft-orthognathic surgery approach than surgery-first noncleft-orthognathic surgery approach, which clinically represents the cleft-associated dental anomalies and transverse maxillary collapse.

The optimal balance between esthetic and functional outcomes, significant reduction in total treatment time, and high rates of patient satisfaction have led to the postulation that the modern surgery-first orthognathic-approach concept may denote a reasonable and cost-effective pathway to manage dentoskeletofacial deformities, and has the potential to become the first-line orthognathic-surgery intervention in the future [18,19,20]. Despite published recommendation for selection of patients for the surgery-first approach [22,23], there are no unique criteria adopted by different centers and clinicians [15,16,17,18,19,20,24,27,28,29,30,31,45,46,47,48,49,50,51,52,53]. While some patterns of presentation (e.g., Class III prognathism with open bite) have been considered as good candidates for the surgery-first approach, a wide spectrum of dental configurations have also been contemplated for surgery-first orthognathic surgery treatment [15,16,17,18,19,20,22,23,24,27,28,29,30,31,45,46,47,48,49,50,51,52,53]. However, patients with cleft-skeletofacial deformities have not been considered as potential candidates to receive this therapeutic modality [24]. This modern approach can theoretically have enhanced influence on patients with clefts who had an extensive dental and orthodontic burden of care since infancy [54]. Due to the prevalence of clefts and the global number of patients requiring orthognathic treatment [1,2,3,4], it is reasonable for this therapeutic option to be considered and investigated.

It is important to emphasize that the surgical-occlusion setup is certainly more technically demanding in cleft than non-cleft deformities due to the complex cleft-related dental abnormalities, such as irregular arch form and shape as well as teeth anomalies [11,55,56]. In the conventional orthodontics-first approach, as presurgical therapy brings maxillary and mandibular teeth into ideal relationships to the underlying skeletal bases, the surgical-occlusion setup is very close to the final occlusion, i.e., the ideal occlusion [11,12,13]. When embracing the surgery-first approach, dental alignment, arch leveling, and coordination, and incisor decompensation are deferred for postsurgical management; the surgical-occlusion setup is consequently different from the final (ideal) occlusion at the end of treatment [22,23,28]. Not only can anteroposterior dental movements be orthodontically adjusted postoperatively, but also transverse and vertical dental movements can be successfully achieved due to the increased metabolic turnover of the regional acceleratory phenomenon [36]. Surgical occlusion in surgery-first treatment was thus set as a treatable malocclusion [22,23,28]. A major concern for this setup is the accurate estimation of the required space for postsurgical dental movement with many combinations of potential directions [22,23,28]. These challenges are probably the major hampering factors for the widespread use of this technique in cleft centers globally.

In this center, the indication of surgery-first orthognathic surgery treatment has been variable [22,23,57,58,59]. In our orthognathic surgery workflow, the combination of accurate clinical examinations and high-quality 3D imaging has permitted a precise preoperative diagnosis that encompasses the many deviations of involvement of the dental, skeletal, and facial soft tissue elements, with less favorable patterns of dental characteristics being not considered contraindication for the surgery-first approach. The rate of indication of this surgery-first protocol has mainly been determined by level of orthodontic experiences to evaluate the accomplishment of a workable surgical-occlusion setup and to anticipate an achievable postoperative orthodontic treatment planning [22,23,57,58,59], with senior experienced professionals (not included as co-authors of this current study) reaching a rate of 100% for surgery-first-based treatments [59]. Therefore, due to the accumulated experience of our team with a high-volume of surgery-first noncleft orthognathic-surgery procedures [22,23,28,32,33,34,35,36,37], the surgery-first approach has progressively been adopted for cleft-skeletofacial deformity. The regular use of virtual simulation has also assisted the change of our cleft practice, as CBCT-derived images allow three-dimensionally appraising the precision of the surgical-occlusion setup in terms of residual or induced skeletal deformity with a designation of surgical feasibility before the actual procedure [40]. For the surgery-first approach, we have indicated the surgical procedure in patients with no need for presurgical therapy or requiring a short period of therapy (standard and modified models, respectively). Other proponents of the surgery-first approach have also adapted models for a short preparatory phase (e.g., “minimal” and “early”), with presurgical therapy ranging 1–6 months, and the preservation of key advantages of the surgery-first pathway (i.e., immediate postoperative improvement of facial appearance with substantial reduction of total treatment time) [45,46,47,48,49,50,51]. In the current study, no differences were found between the cleft and noncleft cohorts for presurgical-therapy time in the modified surgery-first model. Moreover, no need for revisionary surgery was observed during follow-up. These aspects reveal that it is clinically feasible to apply the principles of selection of patient dentition in the cleft scenario with achievement of the desired surgical-occlusion setup, but with no compromise of time and surgical achievability parameters.

As expected, the standard and modified surgery-first models had no significant difference for all tested parameters, reinforcing that the same principles were adopted during the surgical-occlusion setup of the surgery-first approach regardless of a short period of presurgical orthodontic therapy. Logically, patients managed with the modified model had slight differences in dentition status at presentation in our center compared to patients managed by the standard model. Importantly, the main target of the modified surgery-first treatment was not to transform a patient’s dentition with indication for conventional orthodontics-first approach into a favorable dentition to receive a surgery-first approach. Presurgical orthodontics was actually only applied to reduce potential premature contact between maxillary and mandible teeth with the removal of severe occlusal interference enhancing stable surgical occlusion. Minimal decompression of mandibular teeth was also performed when necessary, but with no attempt for decompression of the maxillary teeth.

Different strategies have been employed during surgical-occlusion setup to compensate for space for dental alignment, and arch leveling and coordination after surgery, but with no consensus among advocates of the surgery-first approach and no quantitative data for cleft-related treatment [22,23,27,28,29,30,31,32,33,34,35,36,37,38]. Appraisal of comparative analyses of this study reveals 3D quantitative-based practical fundamentals for surgical-occlusion setup in surgery-first cleft-orthognathic surgery. In the conventional orthodontics-first approach, surgical occlusion was ideally set as a normal overjet (2 mm) and overbite (2 mm) and Class I molar relationship [60]. Because compensation of horizontal mandibular relapse was planned for with a 2 mm overcorrection in surgery-first treatment [28,32], the Class II molar relationship was frequently set in the occlusion of both cleft and noncleft sides. As incisor decompensation was deferred after surgery [28,32], analyses of the cleft cohort exhibited a significantly larger overjet in the surgery-first than the conventional approach. However, the noncleft cohort (mean of 4.37 mm) had a significantly larger overjet than the cleft cohort (mean of 3.31 mm). This is not surprising because the upper incisors of patients with clefts are positioned in a more upright position due to scar contracture from previous surgical interventions [11,55,56].

Definitions of stable occlusion were previously described [15,16,27,30]. To achieve proper tooth contacts with at least three-point teeth contact (preferably one and two at the anterior and bilateral posterior regions, respectively), increasing the posterior open bite by pitch counterclockwise rotation of the distal mandibular segment was formerly recommended [28]. This compromise of superoinferior dental position in the posterior region to attain a better setup in the anterior region was adopted in our cohorts in a case-by-case basis, with the posterior open bite respecting the limit of postoperative orthodontic tooth movement (<10 mm) [22,23,28,32]. Based on the current quantitative data, stable occlusion can be achieved by occlusal contact on one (anterior region), two, or three (most frequent pattern in both cleft and noncleft cohorts) segments or occlusal contact on five to seven teeth, with all of the included patients presenting with at least one point of contact in the anterior maxillary segment. This quantitative pattern of surgical-occlusion setup is similar to a previous report showing quantitative data in noncleft Class III skeletal deformity [28], suggesting that it is possible to achieve stable occlusion even in patients with clefts and associated dental anomalies.

It was advised to not include posterior crossbite at setup [27,31], but our strategies in the transverse dimension emphasized the coordination of jaw midlines instead of the dental arch to avoid positional jaw asymmetry [28]. Current data show that the cleft cohort had significantly smaller values for maxillary transversal-related distances than the noncleft cohort, but with no difference for the mandible region. Transversal deficiencies secondary to scar-tissue contraction are one of the major concerns for professionals treating patients with clefts [61,62,63]. In our center, the rate of maxillary segmental surgery to correct maxillary transversal-related issues has decreased over the years, as selection of patients for each type of procedure, and the technical details have accordingly evolved. Segmental surgery has only been indicated in patients with severe skeletal crossbite. Arch coordination is deferred after surgery with the surgery-first approach, with posterior dental crossbite and mild skeletal crossbite being orthodontically corrected, for example, by bending orthodontic archwire, or inter- or intra-arch elastics.

The potential limitations of this study should be addressed. Due to the adopted study design, we do not provide inter-investigator reliability for quantitative analysis and intra- and inter-professional reliability for occlusion setup, deserving future investigation by using a distinct methodological approach [64,65]. Our results are restricted to a relatively limited number of patients. An a priori sample size and power calculation could not be defined due to the methodological heterogeneity between the current study and prior literature. We also do not provide post-hoc power analysis due to the inadequacy of this particular method [66,67]. In addition, our findings are based on a specific subgroup of young adult patients with unilateral complete cleft lip/palate who were managed by the same orthodontic and surgeon professionals by using two types of orthodontic approaches and a particular surgical technique (3D-assisted single-splint two-jaw procedure). Moreover, patients with variable degree of the dental presentation (curve of Spee, anterior-teeth alignment, incisor inclination, and present dentition) were included in this study. This represents the orthognathic surgery practice in this center and further details of the patient-specific approach have previously been described [22,23,28,32,33,34,35,36,37,39,57,58,59]. This study presented 3D quantitative data derived from the final surgical-occlusion setups of patients who actually received orthognathic treatment and presented no need for revision procedure during follow-up, which denoted satisfactory results. As the patient cohorts were not selected based on surgical results (satisfactory or unsatisfactory), the bias related to analyses performed only on the satisfactory results was considerably reduced, which infers therapeutic feasibility in the cohorts reported here. Moreover, as the applied surgical-orthodontic treatment would considerably vary depending on the dentition status of each patient, orthodontists and surgeons should be aware of the principles and limits of the surgery-first approach during the definition of patient-specific diagnosis and the therapeutic plan (including the prediction of postsurgical change). However, this study does not provide postoperative stability-related statistics, long-term follow up data on results, or information that may guide postoperative arch coordination and dental decompensation, deserving further investigation.

This study did not answer all issues about the addressed topic; however, the current 3D quantitative data can be adopted and adapted by other multidisciplinary teams as initial benchmark values for surgical-occlusion setup in surgery-first cleft-orthognathic surgery. The indication barriers for the surgery-first approach are continuously changing, and we expect that a higher number of patients with clefts would benefit from this modern approach in the future. This may result in changes of the current delivery of cleft-orthognathic surgery care.

## 5. Conclusions

This comparative study of occlusion setup showed: (1) similar 3D quantitative characteristics in standard and modified surgery-first models for the cleft cohort; (2) a smaller number of anterior teeth contacts and larger incisor overjet in the surgery-first cleft-orthognathic surgery approach than the conventional cleft-orthognathic surgery approach; and (3) a larger canine lateral overjet and smaller incisor overjet and maxillary transversal-related distances in the surgery-first cleft-orthognathic surgery approach than surgery-first noncleft-orthognathic surgery approach.

## Figures and Tables

**Figure 1 jcm-08-02116-f001:**
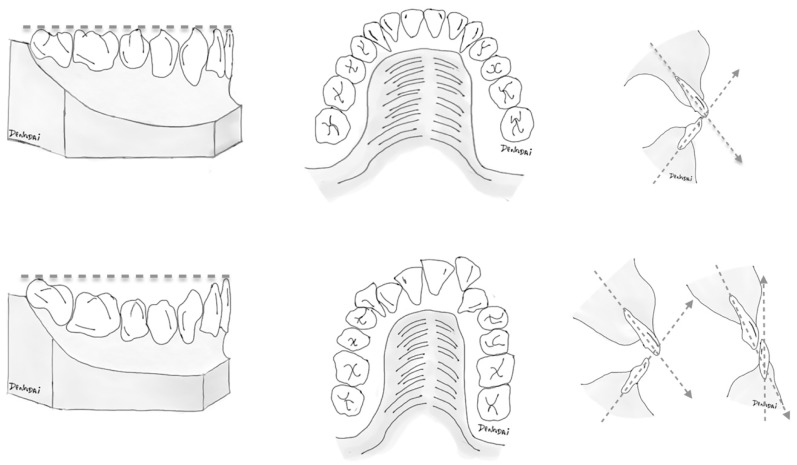
Basic illustrative schemes portraying general dental elements: (**left**) curve of Spee; (**middle**) anterior-teeth alignment; and (**right**) incisor inclination, adopted to distingue (**top**) more favorable (minimal anterior dental crowding, flat-to-mild curve of Spee, and normal range of angle between basal bone and upper and lower incisors) and (**bottom**) less favorable dentition status for management by the surgery-first orthognathic approach. As there is broad clinical presentation with variable degrees of association between these dental elements, patient-specific diagnosis and tailored therapeutic planning should be established in a case-by-case basis.

**Figure 2 jcm-08-02116-f002:**
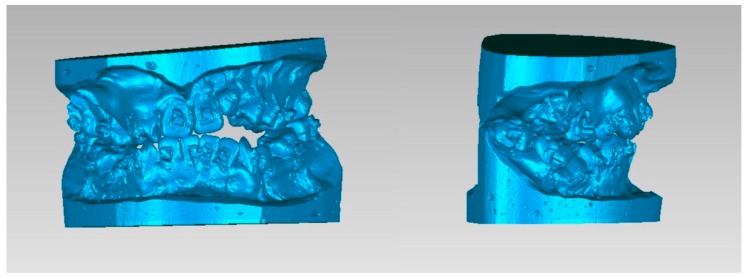
Digitalized dental images displaying final surgical-occlusion setup for surgery-first approach.

**Figure 3 jcm-08-02116-f003:**
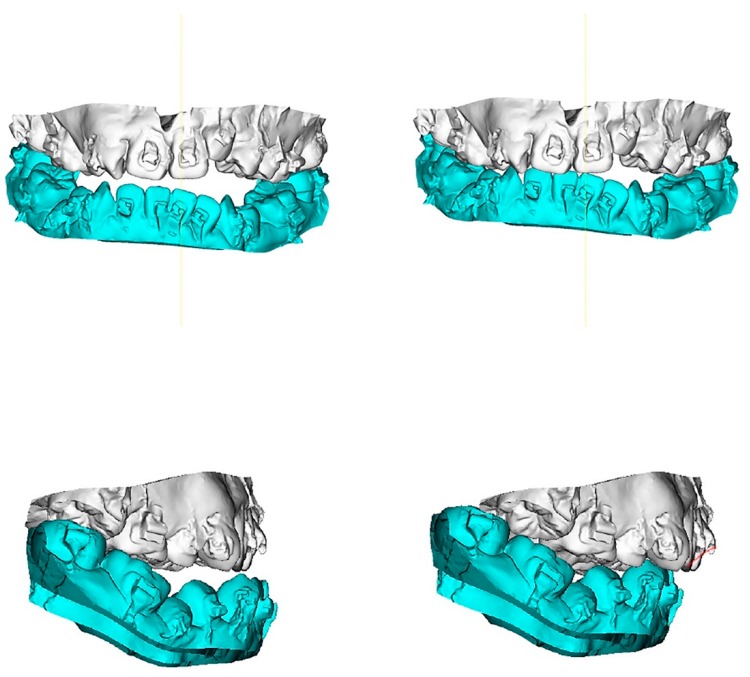
3D maxillary and mandibular dental arch models (**left**) before and (**right**) after mobilization of the osteotomized mandible distal segment to achieve final surgical-occlusion setup. Digitalized dental images displayed in Figure 2.

**Figure 4 jcm-08-02116-f004:**
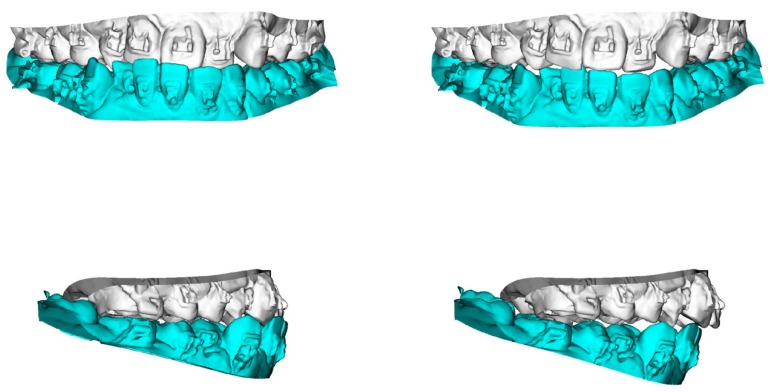
3D maxillary and mandibular dental arch models (**left**) before and (**right**) after mobilization of the osteotomized mandible distal segment to achieve final surgical-occlusion setup.

**Figure 5 jcm-08-02116-f005:**
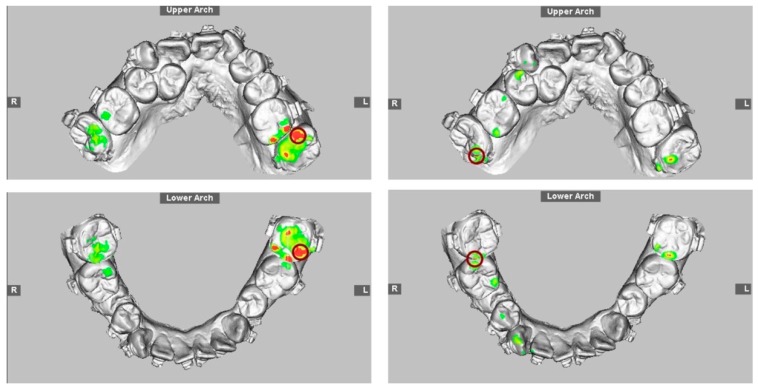
Occlusogram with color map tool displaying 3D (**top**) maxillary and (**bottom**) mandibular dental-arch models (**left**) before and (**right**) after mandible mobilization for occlusion setup, with surgical occlusal contact on three segments and six teeth. Note the reduction of tooth contact in posterior region (red and green color) due to the creation of anterior-tooth contact (green color), which is a characteristic step adopted for surgical-occlusion setup in surgery-first approach. Red indicates degree of (**left**) dental collision, which can be (**right**) thoroughly adjusted before finishing surgical-occlusion setup. For the surgery-first approach, the orthodontic brackets were bonded preoperatively but with no orthodontic tooth movement; wires were placed one day before surgery; tooth #14 was extracted during surgery; orthodontic treatment started during the healing stage by addressing the curve of Spee (a large overjet was designed for the surgery) and the constrict upper posterior teeth with trans-palatal arch appliance. Dental images displayed in Figure 2 and Figure 3.

**Figure 6 jcm-08-02116-f006:**
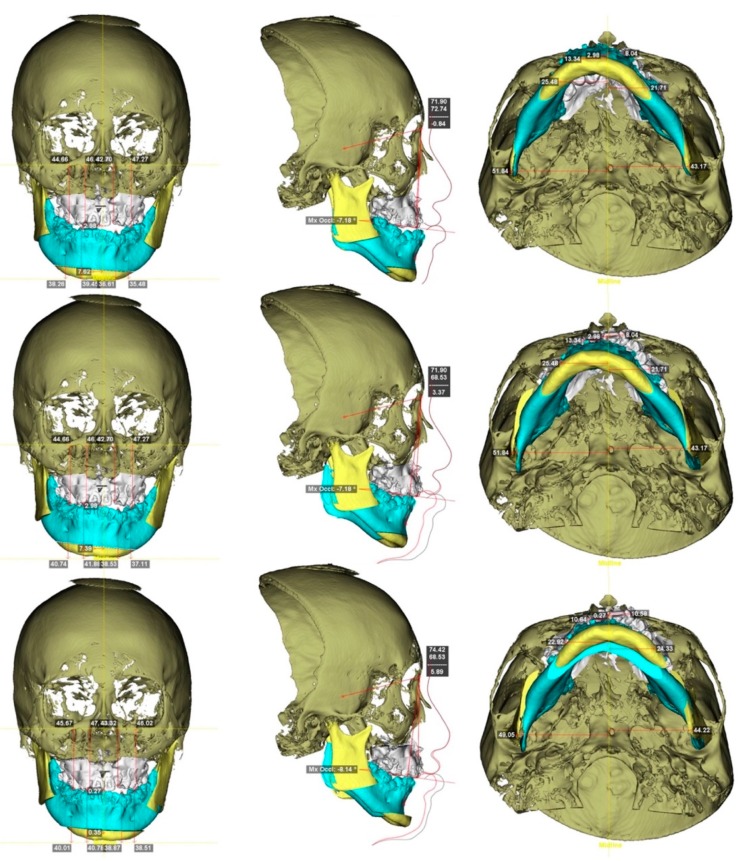
Practical example of 3D computer-assisted single-splint two-jaw surgical simulation using surgery-first approach. (**Top**) Actual cleft–skeletofacial deformity with Le Fort I, bilateral sagittal splint, and genioplasty osteotomy lines. (**Center**) Definition of final surgical-occlusion setup by mobilization of osteotomized distal mandible bone segment. (**Bottom**) Final simulation based on an orthodontic-surgical collaborative decision-making process with the maxillomandibular complex mobilized in translation directions as well as roll, pitch, and yaw rotation movements using frontal, profile, and basal views, respectively. These patient-specific bone mobilizations were accomplished by the maintenance of osteotomized maxilla and distal mandible bone segments as a unique unit (maxillomandibular complex) and with no modification in the final surgical-occlusion setup. This final-simulation dataset was adopted to transfer virtual surgery to actual surgery. Dental images displayed in Figure 2, Figure 3 and Figure 5. Single-splint two-jaw orthognathic surgery technique principle is displayed in Figure 8 and Figure 9.

**Figure 7 jcm-08-02116-f007:**
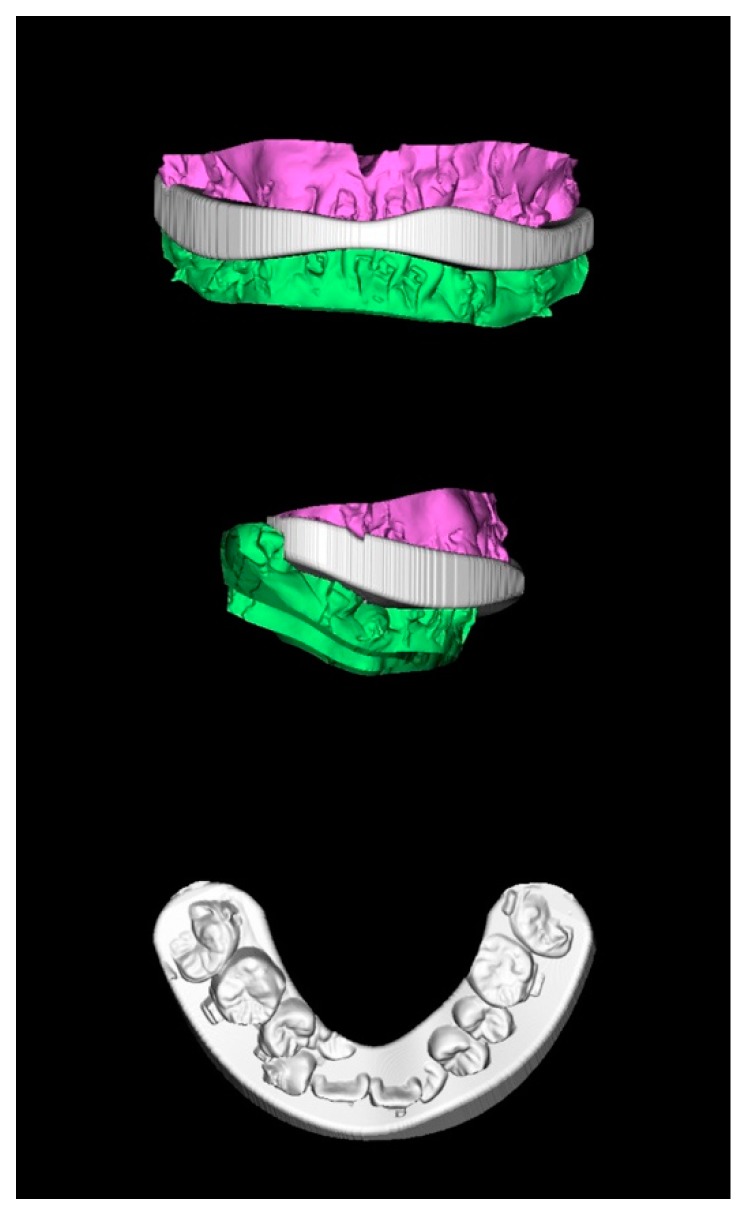
Computer-generated 3D surgical-occlusion splint using surgery-first approach based on final occlusion setup. Dental images displayed in Figure 2, Figure 3, Figure 5 and Figure 6.

**Figure 8 jcm-08-02116-f008:**
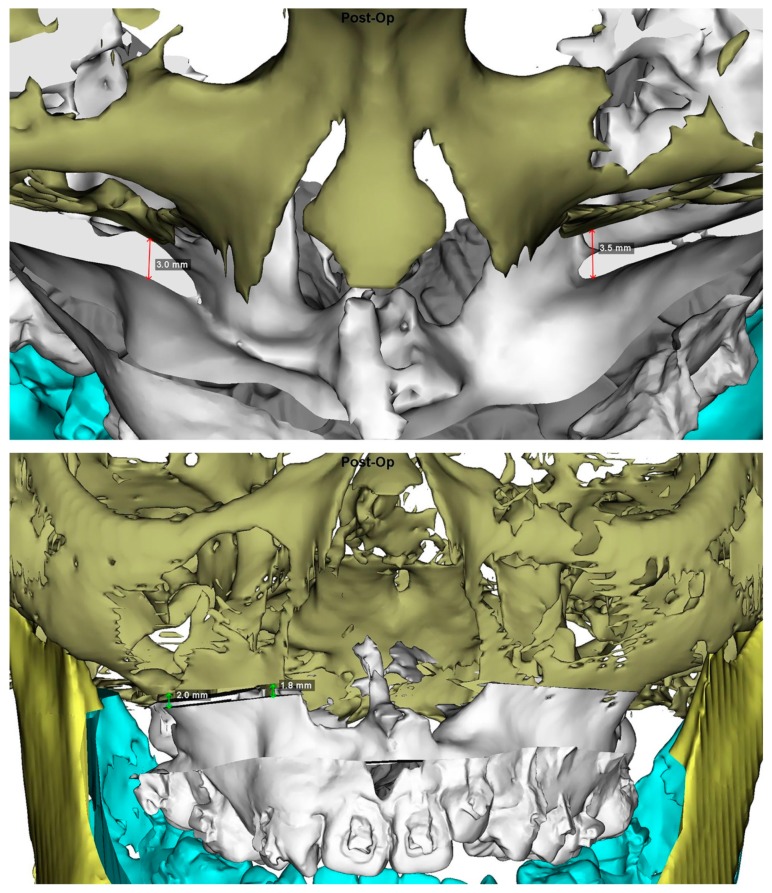
3D imaging views displaying the measurements in medial and lateral maxillary pillars bilaterally which are adopted as guiding template for transferring the 3D planning to actual single-splint technique-based surgical procedure (Figure 9), including (**Top**) advancement in the antero-posterior direction, yaw rotation, (**Bottom**) vertical extrusion, and roll rotation.

**Figure 9 jcm-08-02116-f009:**
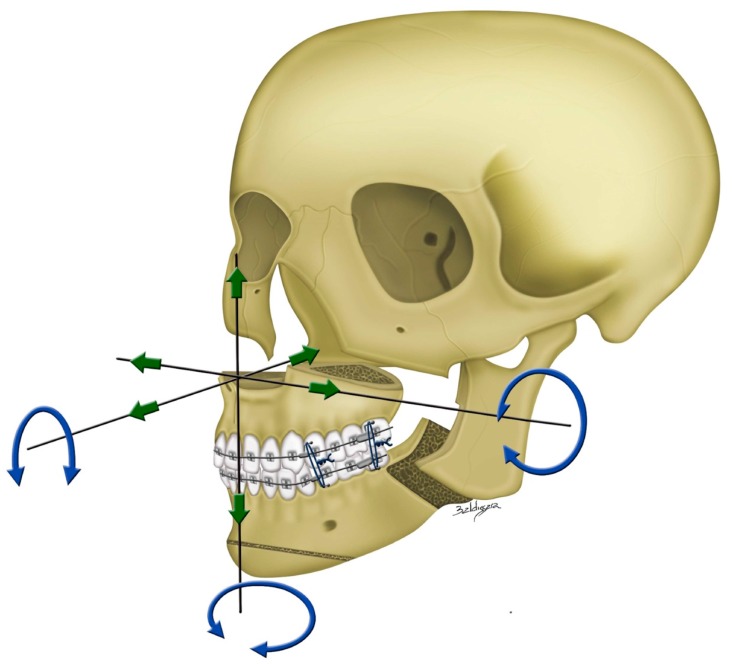
Single-splint two-jaw orthognathic surgery principle. Both maxilla (Le Fort I segment) and mandible (two proximal ramus segments and one distal segment) were completely osteotomized, fixed in the final occlusion using the 3D-printed final surgical-occlusion splint, and moved as an integrated maxillomandibular complex to the 3D-simulated position. To transfer the 3D planning to actual surgery, measurements in maxillary pillars bilaterally (Figure 8), face bow-based midline checking (nasal dorsum and tip, lips, maxilla, dental arches, and chin areas), and middle and lower facial third proportions judgments were used as reference. For this, the maxillomandibular complex was moved in six potential directions, including pitch, roll, and yaw rotations (blue arrows) and en-bloc linear horizontal (left or right shifts and advancements or setbacks in the antero-posterior direction) and vertical (extrusion or intrusion) movements (green arrows).

**Table 1 jcm-08-02116-t001:** Spectrum of dental characteristics in cleft cohort before surgery-first model treatment.

Parameters	Cleft Cohort (*n* = 21)
**Maxillary dentition**	
Missing tooth	
Yes/No *n* (%)	12 (57.1%)/9 (42.9%)
Lateral incisor *n* (%)	12 (57.1%)
Central incisor *n* (%)	1 (4.8%)
First premolar *n* (%)	1 (4.8%)
Spacing	
Yes/No *n* (%)	3 (14.3%)/18 (85.7%)
Anterior crowding	
Yes/No *n* (%)	21 (100%)/0 (0%)
Mild/Moderate/Severe *n* (%)	6 (28.6%)/8 (38.1%)/7 (33.3%)
Posterior crowding	
Yes/No *n* (%)	9 (42.9%)/12 (57.1%)
Mild/Moderate/Severe *n* (%)	5 (55.6%)/2 (22.2%)/2 (22.2%)
**Incisor relation**	
Overjet	
Negative/Positive *n* (%)	21 (100%)/0 (0%)
Negative/Positive (mm) m ± sd	−5.5 ± −3.8/–
Overbite	
Negative/Positive *n* (%)	3 (14.3%)/18 (85.7%)
Negative/Positive (mm) m ± sd	−4.7 ± −1.5/2.6 ± 1.8
**Upper midline deviation**	
Yes/No *n* (%)	17 (81%)/4 (19%)
Yes/No (mm) m ± sd	3.2 ± 1.8/–
**Curve of Spee**	
Cleft side *n* (%)	
Mild/Moderate/Severe	10 (47.6%)/8 (38.1%)/3 (14.3%)
Noncleft side *n* (%)	
Mild/Moderate/Severe	11 (52.4%)/7 (33.3%)/3 (14.3%)
**Posterior crossbite (molar)**	
Cleft side (Yes/No) *n* (%)	9 (42.9%)/12 (57.1%)
Noncleft side (Yes/No) *n* (%)	7 (33.3%)/14 (67.7%)

*n*, number of patients; m, mean; sd, standard deviation; mm, millimeters.

**Table 2 jcm-08-02116-t002:** Comparison of surgical occlusions between standard surgery-first and modified surgery-first models in cleft cohort.

Parameters	Standard Surgery-First Model (*n* = 8)	Modified Surgery-First Model (*n* = 13)	*p*-Value
**Dental-occlusion contacts**			
Number of segmental contacts * *n* (%)			
Three segments	6 (66.7%)	11 (84.6%)	0.425
Two segments	2 (33.3%)	1 (7.7%)	
One segment	0 (0%)	1 (7.7%)	
Number of tooth contacts m ± sd			
Anterior teeth	1.88 ± 0.99	1.69 ± 1.03	0.694
Premolar teeth	2.88 ± 0.83	2.15 ± 2.88	0.139
Molar teeth	2.00 ± 1.41	2.92 ± 1.26	0.135
Total	6.75 ± 1.28	6.77 ± 2.05	0.981
**Incisor midpoint relation** (mm) m ± sd			
Overjet	3.70 ± 2.26	3.07 ± 1.27	0.420
Overbite	1.27 ± 0.90	1.02 ± 0.98	0.574
**Midline deviation** (mm) m ± sd			
Upper midline deviation	0.48 ± 0.53	1.10 ± 1.42	0.254
Midline discrepancy	1.36 ± 1.69	1.10 ± 0.93	0.687
Pogonion deviation	1.37 ± 1.06	1.61 ± 1.46	0.652
**Transverse discrepancy** (mm) m ± sd			
Maxillary intercanine distance	29.70 ± 5.70	29.00 ± 6.45	0.804
Maxillary intermolar distance	49.93 ± 7.40	49.57 ± 4.05	0.888
Mandibular intercanine distance	28.85 ± 2.37	26.54 ± 2.60	0.055
Mandibular intermolar distance	47.55 ± 4.49	46.95 ± 2.54	0.700
Ratio of intercanine distance	1.03 ± 0.22	1.10 ± 0.23	0.535
Ratio of intermolar distance	1.05 ± 0.10	1.06 ± 0.11	0.791
Canine lateral overjet discrepancy	5.97 ± 3.89	4.30 ± 3.00	0.283
First molar lateral overjet discrepancy	3.10 ±1.94	2.24 ± 1.28	0.234
**Skeletal discrepancy** m ± sd			
ANB (°)	3.00 ± 3.90	3.12 ± 3.29	0.937
Advancement of A-point (mm)	3.90 ± 1.53	5.68 ± 2.20	0.059
Setback of B-point (mm)	−4.78 ± 3.59	−5.93 ± 5.01	0.578

*n*, number of patients; m, mean; sd, standard deviation; mm, millimeters; ^o^, degree; ANB, A-point–nasion–B-point angle; A point, point of maximum concavity in midline of alveolar process of maxilla; B point, point of maximum concavity in midline of alveolar process of mandible; *, maxillary arch divided into three segments: anterior, posterior right, and posterior left.

**Table 3 jcm-08-02116-t003:** Comparison of surgical occlusions between the surgery-first and conventional approaches in cleft cohort.

Parameters	Surgery-First Approach (*n* = 21)	Conventional Approach (*n* = 23)	*p*-Value
**Dental occlusion contacts**			
Number of segmental contacts * *n* (%)			
Three segments	17 (81.0)	18 (78.3)	0.873
Two segments	3 (14.3)	3 (13.0)	
One segment	1 (4.7)	2 (8.7)	
Number of teeth contacts m ± sd			
Anterior teeth	1.76 ± 1.00	2.52 ± 1.38	0.044
Premolar teeth	2.42 ± 1.08	2.57 ± 1.04	0.670
Molar teeth	2.57 ± 1.36	2.04 ± 1.26	0.189
Total	6.76 ± 1.76	7.13 ± 2.30	0.557
**Incisor midpoint relation** (mm) m ± sd			
Overjet	3.31 ± 1.69	2.12 ± 1.26	0.011
Overbite	1.11 ± 0.93	0.51 ± 1.25	0.077
**Midline deviation** (mm) m ± sd			
Upper midline deviation	0.86 ± 1.18	0.83 ± 1.03	0.923
Midline discrepancy	1.21 ± 1.43	1.02 ± 1.18	0.634
Pogonion deviation	1.52 ± 1.30	1.44 ± 1.01	0.826
**Transverse discrepancy** (mm) m ± sd			
Maxillary intercanine distance	29.27 ± 6.04	30.50 ± 5.60	0.485
Maxillary intermolar distance	49.70 ± 5.39	50.10 ± 3.69	0.774
Mandibular intercanine distance	27.42 ± 2.71	27.73 ± 1.31	0.625
Mandibular intermolar distance	47.18 ± 3.32	46.77 ± 3.39	0.687
Ratio of intercanine distance	1.07 ± 0.22	1.10 ± 0.22	0.646
Ratio of intermolar distance	1.06 ± 0.11	1.08 ± 0.09	0.515
Canine lateral overjet discrepancy	4.94 ± 3.37	3.60 ± 2.65	0.149
First molar lateral overjet discrepancy	2.57 ± 1.58	1.78 ± 1.34	0.080
**Skeletal discrepancy** m ± sd			
ANB (^o^)	3.41 ± 3.55	2.54 ± 2.32	0.337
Advancement of A-point (mm)	5.00 ± 2.12	5.00 ± 1.84	0.997
Setback of B-point (mm)	5.49 ± 4.46	6.06 ± 5.06	0.698

*n*, number of patients; m, mean; sd, standard deviation; mm, millimeters; ^o^, degree; ANB, A-point–nasion–B-point angle; A point, point of maximum concavity in midline of alveolar process of maxilla; B point, point of maximum concavity in midline of alveolar process of mandible; *, maxillary arch divided into three segments: anterior, posterior right, and posterior left.

**Table 4 jcm-08-02116-t004:** Comparison of surgery-first and conventional approaches in cleft cohort.

	Cleft Side			Noncleft Side		
Parameters	Surgery-First Approach (*n* = 21)	Conventional Approach (*n* = 23)	*p*-Value	Surgery-First Approach (*n* = 21)	Conventional Approach (*n* = 23)	*p*-Value
**Dental occlusion contact**						
Angle classification *n* (%)						
Angle Class I	2 (9.6)	1 (4.3)	0.496	1 (4.8)	4 (17.4)	0.257
Angle Class II	19 (90.4)	22 (95.7)		19 (90.4)	19 (82.6)	
Angle Class III	0 (0)	0 (0)		1 (4.8)	0 (0)	
Number of tooth contact m ± sd						
Anterior teeth	0.76 ± 0.70	1.22 ± 0.67	0.033	1.00 ± 0.55	1.30 ± 1.11	0.249
Premolar teeth	1.24 ± 0.62	1.35 ± 0.65	0.571	1.19 ± 0.68	1.22 ± 0.67	0.896
Molar teeth	1.24 ± 0.89	1.00 ± 0.74	0.338	1.33 ± 0.86	1.04 ± 0.82	0.259
Total	3.24 ± 1.14	3.57 ± 1.34	0.390	3.52 ± 1.17	3.57 ± 1.41	0.916
**Dental inclination** (^o^) m ± sd						
U1 inclination	98.37 ± 14.27	105.96 ± 9.73	0.044	101.72 ± 9.54	106.61 ± 8.13	0.074
Interincisal angle	135.20 ± 15.22	123.38 ± 12.64	0.007	130.64 ± 12.25	122.96 ± 11.36	0.037
**Anteroposterior relation** (mm) m ± sd						
U1 overjet	3.24 ± 2.12	1.82 ± 1.27	0.009	3.39 ± 1.43	2.42 ± 1.44	0.030
U6 overjet	6.41 ± 3.85	7.16 ± 5.73	0.609	3.65 ± 4.27	4.29 ± 4.43	0.628
**Vertical relation of teeth**						
U1 overbite (mm) m ± sd	0.80 ± 1.18	0.24 ± 1.31	0.144	1.43 ± 0.88	0.77 ± 1.29	0.057
Posterior open bite (U7) *n* (%)						
Yes	11 (52.4)	6 (26.1)	0.041	13 (61.9)	7 (30.4)	0.035
No	10 (47.6)	17 (73.9)		8 (38.1)	16 (69.6)	
**Transverse arch coordination** m ± sd						
Midline to U3 distance (mm)	12.66 ± 3.41	14.02 ± 3.15	0.177	16.19 ± 3.57	16.18 ± 3.25	0.996
Midline to L3 distance (mm)	13.74 ± 2.89	14.03 ± 1.40	0.667	13.61 ± 2.62	13.89 ± 3.10	0.749
Midline to canine ratio	0.95 ± 0.29	1.02 ± 0.29	0.470	1.20 ± 0.24	1.17 ± 0.25	0.640
Canine lateral overjet (mm)	−1.08 ± 3.79	−0.01 ± 3.97	0.368	2.58 ± 2.94	2.29 ± 3.84	0.785
Midline to U6 distance (mm)	25.08 ± 2.97	24.89 ± 2.48	0.811	24.96 ± 3.73	25.18 ± 2.27	0.813
Midline to L6 distance (mm)	23.57 ± 2.88	23.49 ± 2.31	0.918	23.05 ± 2.27	23.11 ± 2.36	0.933
Midline to first molar ratio	1.07 ± 0.12	1.06 ± 0.11	0.838	1.08 ± 0.14	1.10 ± 0.09	0.749
First molar lateral overjet	1.52 ± 2.82	1.40 ± 2.43	0.883	1.91 ± 3.16	2.07±1.98	0.838
**Maxillary occlusion angle** (^o^) m ± sd	−9.08 ± 7.27	−11.79 ± 3.91	0.139	−10.45 ± 5.51	−10.99 ± 4.65	0.726
**Jaw movement** (mm) m ± sd						
Advancement of U3	4.12 ± 2.98	4.45 ± 2.54	0.698	3.49 ± 3.29	2.87 ± 2.20	0.467
Impaction of U3	0.99 ± 2.40	0.50 ± 1.68	0.436	0.32 ± 2.02	0.22 ± 2.02	0.872
Advancement of U6	4.43 ± 3.07	5.08 ± 3.08	0.492	3.34 ± 3.09	2.37 ± 2.73	0.277
Impaction of U6	−0.50 ± 1.31	−0.49 ± 1.67	0.980	−1.05 ± 1.70	−1.12 ± 2.35	0.910
Setback of mandible	−4.35 ± 4.73	−4.90 ± 5.40	0.722	−4.64 ± 4.18	−6.24 ± 3.97	0.199

U1, upper central incisor; U3, upper canine; L3, lower canine; U6, upper first molar; L6, lower first molar; U7, upper second molar.

**Table 5 jcm-08-02116-t005:** Comparison of cleft and noncleft cohorts.

Parameters	Cleft Cohort (*n* = 21)	Noncleft Cohort (*n* = 18)	*p*-Value
**Dental-occlusion contact**			
Number of segmental contacts * *n* (%)			
Three segments	17 (81.0)	15 (83.3)	0.638
Two segments	3 (14.3)	3 (16.7)	
One segment	1 (4.7)	0 (0)	
Number of teeth contacts m ± sd			
Anterior teeth	1.76 ± 1.00	2.50 ± 1.38	0.061
Premolar teeth	2.42 ± 1.08	2.22 ± 1.44	0.612
Molar teeth	2.57 ± 1.36	2.72 ± 1.18	0.716
Total	6.76 ± 1.76	7.44 ± 2.23	0.292
**Incisor midpoint relation** (mm) m ± sd			
Overjet	3.31 ± 1.69	4.37 ± 1.27	0.036
Overbite	1.11 ± 0.93	1.46 ± 1.62	0.432
**Midline deviation** (mm) m ± sd			
Upper midline deviation	0.86 ± 1.18	0.55 ± 0.42	0.287
Midline discrepancy	1.21 ± 1.43	0.54 ± 0.67	0.063
Pogonion deviation	1.52 ± 1.30	1.38 ± 1.46	0.749
**Transverse discrepancy** (mm) m ± sd			
Maxillary intercanine distance	29.27 ± 6.04	36.04 ± 2.67	<0.001
Maxillary intermolar distance	49.70 ± 5.39	54.61 ± 3.55	0.002
Mandibular intercanine distance	27.42 ± 2.71	27.12 ± 3.33	0.756
Mandibular intermolar distance	47.18 ± 3.32	47.31 ± 3.05	0.900
Ratio of intercanine distance	1.07 ± 0.22	1.34 ± 0.16	<0.001
Ratio of intermolar distance	1.06 ± 0.11	1.16 ± 0.10	0.004
Canine lateral overjet	4.94 ± 3.37	1.37 ± 1.35	<0.001
First molar lateral overjet	2.57 ± 1.58	2.30 ± 1.84	0.626
**Skeletal discrepancy** m ± sd			
ANB (^o^)	3.41 ± 3.55	3.28 ±1.19	0.882
Advancement of A-point (mm)	5.00 ± 2.12	3.86 ± 1.52	0.066
Setback of B-point (mm)	−5.49 ± 4.46	−6.23 ± 3.34	0.567

*n*, number of patients; m, mean; sd, standard deviation; mm, millimeters; ^o^, degree; ANB, A-point–nasion–B-point angle; A point, point of maximum concavity in midline of alveolar process of maxilla; B point, point of maximum concavity in midline of alveolar process of mandible; *, maxillary arch divided into three segments: anterior, posterior right, and posterior left.

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
