# Peer review of "Modern Surgery-First Approach Concept in Cleft-Orthognathic Surgery: A Comparative Cohort Study with 3D Quantitative Analysis of Surgical-Occlusion Setup"

_jcm, 2019, doi:10.3390/jcm8122116_

Round 1
Reviewer 1 Report
Thank you for the additions /changes in your paper. It is clearly understandable now.
You should definitely follow up your patients to be able to prove that also long term stable results can be achieved with your approach to orthognathic surgery in cleft cases.
Reviewer 2 Report
GUIDELINES: I suggest that the article should be compliant with STROBE guidelines
https://www.strobe-statement.org/index.php?id=strobe-home
TITLE: please, indicate the type of the study in the title (cohort study)
INTRODUCTION:
Overall, I think that the introduction should be more focused on the benefits of a surgery-first approach in cleft lip and palate patients. I suggest to pointed out that cleft lip and palate patients have a significant aesthetic impairment (Staderini E, et al. Lay People Esthetic Evaluation of Primary Surgical Repair on Three-Dimensional Images of Cleft Lip and Palate Patients. Medicina (Kaunas). 2019 Sep 8;55(9). pii: E576. doi: 10.3390/medicina55090576) and the main benefit of the surgery-first approach is an immediate ingrease of the quality of life after surgery, due to the improvement of facial appearance (Pelo S, et al. Surgery-first orthognathic approach vs traditional orthognathic approach: Oral health-related quality of life assessed with 2 questionnaires. Am J Orthod Dentofacial Orthop. 2017 Aug;152(2):250-254. doi: 10.1016/j.ajodo.2016.12.022.). Moreover, the rate of complications related to surgery first approach may be slightly higher than those associated with traditional orthognathic surgery (Pelo S, et al. Risks in surgery-first orthognathic approach: complications of segmental osteotomies of the jaws. A systematic review. Eur Rev Med Pharmacol Sci. 2017 Jan;21(1):4-12.).
Line 52: I think that the number of references (six) should be limited to the most relevant ones.
The aim of the study should be indicated with primary (and secondary) end-points with a null hypothesis.
MATERIAL AND METHODS:
line 92: please, can you change "3D image datasets" with "CBCT scan"?
line 177-121:"are there specific evidence-based guidelines for judging the dental conditions for a surgery-first treatment?" you can read the paper of Kwon and Han (Kwon TG, Han MD. Current status of surgery first approach (part II): precautions and complications. Maxillofac Plast Reconstr Surg. 2019 Jun 3;41(1):23. doi: 10.1186/s40902-019-0206-4.), that state:"From early papers on the SFA, Baek et al. [24] claimed that the SFA is indicated when there is only little or no transverse discrepancy, no extractions involved, and at least three occlusal contact points between the arches. They also suggested that mild to moderate curve of Spee or vertical problem could also be acceptable for the SFA.
Liou et al. [19, 25] mentioned that normal to mildly proclined/retroclined incisor inclination could be permissible in the SFA. These articles implied that cases with severe proclined/retroclined incisors or vertical problems would be the contraindication for the SFA. Later, the “inferior subapical osteotomy” has been proposed to surgically decompensate the severely retroclined mandibular incisors [26]. This also illustrates the changes in the indications and contraindications with the evolution of technique and clinical experience."
Sample size calculation is missing.
Technical error of the measurement (intra- and inter-operator reliability) is missing. Measurements were performed by one investigator.
RESULTS:
Did you observe any complication after surgery?
DISCUSSION:
It should be focused on the interpretation of the results rather than occlusal stability, as this study did not compare post-surgical 3D image records, but only final surgical-occlusion setup
I think this paragraph should be focused on the interpretation of the resut
Limitations and strenghts of the study are missing.
line 373: I think that the number of references (more than ten) should be limited to the most relevant ones.
CONCLUSION:
it should be consistent with primary endpoints and the results
Round 2
Reviewer 2 Report
The article has been significantly improved.
Small suggestions below:
INTRODUCTION
I would write that the primary endpoint of the study is the comparison of surgical-occlusion setup between surgery-first (experimental group) and conventional orthognathic surgery (control group) in cleft patients/cohort.
As i see in the conclusion, you add a secondary endpoint, that is the comparison of surgical-occlusion setup between cleft and noncleft patients/cohorts.
Null hypothesis:
No difference of surgical-occlusion setup exists between surgery-first (experimental group) and conventional orthognathic surgery (control group) in cleft patients/cohort.
No difference of surgical-occlusion setup exists between cleft and noncleft patients/cohorts.
METHODS:
line 131: it seems that a sentence is missing (there is a colon at the end of the sentence)
line 293: please, change "1-month" with "one-month"
line 294-295: the ICC values shoud be moved in the results section
I can't find (except for Table 1 for cleft patients) how many patients did surgery-first and conventional approach. Can you describe it in the materials and methods, please?
RESULTS:
Please, try to organize the data with two sub-sections: primary endpoint (comparison of conv and surg-first in cleft cohort) and secondary endpoint (comparison of cleft and noncleft cohorts).
The tables should report data following the abovementioned scheme; the title of the table 4 should be the same of the primary endpoint, while the title of the table 5 should be the same of the secondary endpoint.
I would move section 3.1 before section 2.5.
Can you change the "standard and modified" with "surgery-first and conventional"? The more consistent the name of approaches are, the more the study gains readability.
Table 4: the title of the table 4 should be the same of the primary endpoint. In table 4, if you compare cleft and non-noncleft sides in surgery-first approach, it looks that interincisal angle is lower but U1 inclination is higher. Are there any typos, or it is related to moderate tooth crowding in the lower arch? Moreover, if you look the cleft side, it seems that conventional orthognathic surgery achieves a significantly less overjet in cleft side, and there is a significanty higher number of anterior contacts. Do you agree?
I think that table 5 should adopt the same values of table 4.
DISCUSSION:
the first part of this section should be focused on data interpretation, and it should be divided in two sub-sections (primary and secondary endpoints). For example, try to give a reason why cleft patients have smaller overjet in cleft side with conventional orthognathic surgery than surgery first, and consequently higher anterior contacts. You should also explain why this finding is relevant for the clinician.
CONCLUSION:
I think the summary should contains all the statistically significant findings that could be interesting for the clinician.
Author Response
Please see the attachment

This manuscript is a resubmission of an earlier submission. The following is a list of the peer review reports and author responses from that submission.
Round 1
Reviewer 1 Report
I am not a specialist in orthognathic surgery or cleft treatment. Do have extensive experience in assessing experimental design and in 3D quantitative assessment of the craniofacial complex.
To the extent that my experience allows, I assess this paper to be well thought through, executed and written and suggest only some minor typographical revisions.
Ln 43 ‘cleft stigmata-related’ -> cleft stigma-related
Ln 99 ‘orthodontics (BCJP)’ -> orthodontist (BCJP)
Ln 146 It isn’t clear to me what ‘the 3D skull models were re-oriented according to clinical measurements, cranial symmetry, and the Frankfurt horizontal plane ’ means
Ln 205 ‘distribution of the data was verified’ -> ‘It was verified that the data were normally distributed’
Reviewer 2 Report
The research on this paper is based on an area of great interest and I found the topic of the paper very existing and was looking forward to the data set which had the potential to be a very rich and give a significant insight on to the surgery first approach in cleft lip and/or palate patients. However, I found the paper to be very confusing and hard to understand with serious reservations on the general technical soundness of this study and robustness of the presented data.
The Topic - Nochanges necessary
Abstract - Minorchanges suggested to improve clarity.
The abstract of this paper has provided concise and thorough summary of the content of the paper but however naming convention of the second study group “developmental dentofacial deformity” group gives a wrong impression if the abstract is referred to as standalone without reference to the main text which is detailed later.
Introduction
This has provided sufficient background information for the reader and provided a brief insight of the gap in the existing knowledge and the importance of the research topic. The authors have clearly defined the aims and objectives of the study.
However, in line 52 and 53 the statement on worsening of facial appearance with presurgical orthodontics is agreeable in non-cleft class III patients, in cleft lip and palate class III patients it is the reviewers view that such generalization is not appropriate. In cleft lip/palate patients with skeletal and class III patients presurgical orthodontics are mainly concerned with arch coordination and crowding rather than decompensation. Specially in the maxillary arch and incisors are not required significant decompensation to which the authors agree to in line 337 and 338. Also, it is generally not need ed to decompensate the mandibular incisors to which the authors agree too on line 112. Therefore, reviewer feels that this argument on worsening of facial appearance is invalid in the cleft cohort. In contrary orthodontic treatment at young age will greatly benefit the cleft lip/palate patients in improvement of general dental/oral health.
As this study is centered on dental occlusion and since there is no universal agreement on terms like occlusion and surgical occlusion it would be best to define the authors interpretation of the terms for clarity and interpretation of the results.
Material, Methods and Results.
This was a retrospective study. The study groups are not clearly defined at the beginning. This structure can only be understood at looking at the results section. These groups must clear be described under material/sampling.
Even though the inclusion and exclusion criteria have been mentioned later in line 96, 98,99, 366 and 367 it is mentioned the retrieved records are done by the same surgeon and the same orthodontist. I believe it should be included in the inclusion/exclusion criteria.
The term in the inclusion criteria “skeletally matured” is vague. It is interesting to know the age range of the cohorts and also the method employed to determine the the skeletal maturity in this retrospective data set.
It is my impression that developmental cohort denotes non cleft, non syndromic adults patients with skeletal class III whose growth has stopped. It would be best to change the labeling of the group to less ambiguous label.
Further it is said in line 105 and 106 the study group that was taken for surgery 1stapproach which is further illustrated in figure 01. The reviewer completely agrees with the statement and consider them good candidates for the surgery first approach. However, in the provided illustration in figure 04 it is the opinion of the reviewer that this represent a case of severe anterior crowding with missing laterals. Therefore, this raises the doubt about the selected data as this should clearly be not included in the surgery 1stapproach according to the authors criteria set in line 105 and 106. This is compounded by the fact that the authors have not provided any data on the pretreatment dental/occlusal clinical features of the surgery 1st(or the conventional group) such as present dentition, incisor relationship, molar relationship, crowding, over jet, overbite etc which would help the reader to draw a picture of the surgery 1stgroup specially in the backdrop of the provided figures.
Also this begs the following questions regarding the figure 04.
How was the crowding was relieved after surgery in this particular case? (Extraction? secondary surgery procedure?) Is it the common practice of the authors to keep cleft patients with class III severe crowding and missing teeth until they become skeletally matured? (The ages ?????)
The overall impression of this retrospective study is written in a manner that gives a impression this was a prospective study.
Although this paper has looked into an area of great interest and merit, as it stands, I believe there are several serious general and specific flaws of this paper from readability, sampling, study design, and reporting of the data and therefore I suggest rejecting this paper. I sincerely hope my critiques will aid the authors in the future to design a stronger study and present this data in a refined manuscript.
Reviewer 3 Report
This is a well designed study but still has a rather small number of patients enrolled. It may be seen as a pilot study. A long term follow up of the results (at least 1-2 years ) needs to be included.
As for the operation you always refer to a 3D-printed final occlusion splint in a two- jaw surgery. There must be an intermediary splint, however, to fix the maxilla first before you can split the mandible. Otherwise you have no orientation as for the position of the maxillo-mandibulary complex in spite of all computer based 3D planning. This point has definitely to be clarified.
The lines127-129 are repeated in the lines 138-140 and have to be taken out.
Please add the ethical approval of your institution.
Round 2
Reviewer 2 Report
I thank the effort of the authors in revising the manuscript substantially which has greatly increased the
understanding of the paper and general readability.
However, to understand the outcome of this study, the data of the population studied must be presented in detail. I believe for the reader, the clinical features, specially the dental clinical parameters (Over-jet, crowding/spacing, dentition etc.) is mandatory to arrive at meaningful conclusion in this study. From the provided limited data in the inclusion criteria and in figure 01, and looking at the example provided in figure 05, it is my opinion that this is not compatible with the intended study population for surgery first group and there is significant bias present.
Therefore, I believe there are several serious general and specific flaws of this paper from in sampling, study design, and reporting of the data which is not sufficiently revised for publication. However, I encourage the authors to re-sample the study and publish them for the benefit of the scientific community.
Reviewer 3 Report
Thank you for this second version in which you clarified a lot of points.
To do without an intermediary splint and completely rely on your 3D measurements of the new position of the mandibulo-maxillary block may definitely be possible but needs completely intact maxillary walls after osteotomy and mobilization. Especially in difficult cleft cases this can not always be guaranteed. Be aware that there may be some letters to the editor regarding this point after the publication of the paper.